# Water Quality Measurement and Modelling Based on Deep Learning Techniques: Case Study for the Parameter of Secchi Disk

**DOI:** 10.3390/s22145399

**Published:** 2022-07-20

**Authors:** Feng Lin, Libo Gan, Qiannan Jin, Aiju You, Lei Hua

**Affiliations:** 1College of Electrical Engineering, Zhejiang University, Hangzhou 310027, China; 22110093@zju.edu.cn; 2Zhejiang Institute of Hydraulics and Estuary, Hangzhou 310020, China; jqn.amy@126.com (Q.J.); yajyaj1997@126.com (A.Y.); hualei@hhu.edu.cn (L.H.)

**Keywords:** Secchi disk, deep learning, transparency, water quality measurement, image processing

## Abstract

The Secchi disk is often used to monitor the transparency of water. However, the results of personal measurement are easily affected by subjective experience and objective environment, and it is time-consuming. With the rapid development of computer technology, using image processing technology is more objective and accurate than personal observation. A transparency measurement algorithm is proposed by combining deep learning, image processing technology, and Secchi disk measurement. The white part of the Secchi disk is cropped by image processing. The classification network based on resnet18 is applied to classify the segmentation results and determine the critical position of the Secchi disk. Then, the semantic segmentation network Deeplabv3+ is used to segment the corresponding water gauge at this position, and subsequently segment the characters on the water gauge. The segmentation results are classified by the classification network based on resnet18. Finally, the transparency value is calculated according to the segmentation and classification results. The results from this algorithm are more accurate and objective than that of personal observation. The experiments show the effectiveness of this algorithm.

## 1. Introduction

Water is an important natural resource for all life. However, due to the rapid development of industrial and agricultural production activities and urbanization, water pollution is becoming more and more serious, directly affecting daily life and production [1,2,3,4]. Some indicators are needed to evaluate the quality of water, one of which is the transparency of water [5,6]. Transparency is defined as the degree to which light penetrates a water body [7]. It plays an important role in many fields such as ecological management [8,9], aquaculture [10], primary production of phytoplankton [11], seagrasses health [12], coral reefs [13] and so on. Therefore, it is of great significance to obtain accurate transparency values for guiding daily factory manufacturing and life.

At present, the commonly used method to measure transparency is the Secchi disk method [14]. It is a checkered black-and-white disk. When in use, it is immersed in water and slowly sinks until the white part of the disk disappears [15,16], and the depth of the disk in the water is the value of transparency. Although it is relatively simple to operate and easy to carry, the accurate critical position of the Secchi disk is not easy to observe with the naked eye. The measurer usually needs to spend several minutes to observe the Secchi disk close to the critical position. In order to obtain accurate results, many people often need to observe it many times, which is time-consuming. Moreover, the results from personal measurement are readily affected by subjective and objective factors, such as the quality of human vision, the experience of operating the Secchi disk, the strength of the surrounding light, the shaking of the water gauge, and so on [17,18]. Therefore, personal measurement using the Secchi disk has great uncertainty and instability [19,20,21], and the Secchi disk method needs to be further improved.

In addition to the method of using a Secchi disk, there are some more advanced methods, which are mainly divided into two categories: using sensor and image processing technology. The turbidity sensor is used to measure the transparency of a body of water [22,23], resulting in the automatic measurement of transparency, but the design, manufacture, and maintenance of a turbidity sensor need a certain cost. At present, the popular image measurement method is used to measure the transparency of water quality through satellite spectrogram. The relationship between satellite spectral images and water transparency can be described by algorithms, and these algorithms can be classified into empirical algorithms, semi-analytical algorithms, analytical algorithms, or machine learning algorithms [24,25]. An empirical algorithm is based on the observation that there is a strong relationship between water transparency and parameters such as reflectance, water-leaving radiances, diffuse attenuation coefficients, and so on, and these parameters can be obtained from satellite spectral images [26]. Semi-analytical algorithms and analytical algorithms are based on the theories of underwater light radiative transmission [27], and they calculate the absorption coefficient and scattering coefficient of the water body through remote-sensing reflectance and construct an equation between these parameters and the water transparency [28,29]. Machine learning algorithms are mentioned in the next paragraph; however, this method is suitable for large areas and needs satellites to obtain image data.

With the development of artificial intelligence, machine learning is used to monitor and assess water parameters such as dissolved oxygen, chlorophyll-a, Secchi disk depth, and so on. In the last decade, an Artificial Neural Network (ANN) is one of the most utilized artificial intelligence methods [30]. Sentas et al. [31] used three models, including ANN, to forecast daily dissolved oxygen. Karamoutsou and Psilovikos [32] used chlorophyl-a, pH, water temperature, water conductivity, turbidity, ammonia nitrogen, and nitrate nitrogen as inputs for the Deep Neural Network (DNN) to predict the dissolved oxygen. Gómez et al. [33] combined satellite images with Random Forest (RF), Support Vector Machine (SVM), ANN, and DNN to monitor chlorophyll-a. Some studies also use ANN for Secchi disk depth measurement. Heddam [34] collected other parameters (total suspended solids, water temperature, dissolved oxygen, and chlorophyll) of the target water area and used ANN to predict the Secchi disk depth. Batur et al. [35] and Arias-Rodriguez et al. [36] combined satellite images with machine learning methods to predict the Secchi disk depth. In recent years, deep learning has made great progress and has more advantages than traditional image processing technology in target detection [37,38], semantic segmentation [39], and so on. Oga et al. [40] and Montassar et al. [41] used semantic segmentation and convolutional neural networks (CNN) to evaluate the turbidity of the target water body, which indirectly reflected the clarity of the water body but failed to measure the specific value of transparency.

Although the above methods are better than the traditional Secchi disk method in different ways, there are few studies that combine deep learning with the Secchi disk. Now, cameras can be seen everywhere, making it easier and cheaper to obtain RGB video. If water transparency can be obtained directly from the Secchi disk video, it will be very worthwhile. When using deep learning to obtain transparency from Secchi disk video, there are two problems, including how to detect the blurred Secchi disk and how to measure the depth of the Secchi disk. For the first problem, some studies use deep learning to detect the blurred object; for example, Wang et al. [42] used DNN, and Zeng et al. [43] used a method based on CNN. For the second problem, a water gauge is commonly used to measure the depth of the Secchi disk. Some studies use deep learning methods to recognize the water gauge. Lin et al. [44] used a semantic segmentation network to segment the water gauge and processed the segmentation result to calculate its reading. Wang et al. [45] used a DNN to recognize the ship water gauge and calculated its reading according to the recognition result. Based on existing studies, it is possible to measure transparency by combining deep learning with the Secchi disk.

In this paper, image processing techniques and deep learning are combined with the Secchi disk method to measure the transparency value of water. A general camera (VARID-SUMMILUX-H) is used to take a video of the measurement process of the Secchi disk method. The critical position of the Secchi disk and the corresponding water gauge value are determined by image processing of the Secchi disk video and water gauge video, respectively. A Faster RCNN [46] is applied to determine the critical position of the Secchi disk, and the white part of the disk is segmented from the image by using the OTSU algorithm (Maximization of interclass variance), and the resnet18 [47] network is used to classify the segmentation results; furthermore, the critical position of the Secchi disk was determined according to the classification results. For the acquisition of the water gauge scale, firstly, the semantic segmentation network Deeplabv3+ [48] was used to segment the corresponding water gauge, and then binary and k-means clustering operations were performed on the segmented water gauge. Next, the characters were segmented, and the characters were classified by the resnet18 network. Finally, the scale value of the water gauge was calculated. When the Secchi disk is at the critical position, the corresponding water gauge scale is the transparency value.

Basically, the main contributions and novelty of this work are as follows: creatively proposes an algorithm based on deep learning and image processing technology to measure the transparency of water quality with Secchi disk. The algorithm adopts a method to determine the critical position of the Secchi disk, which can accurately determine the critical position of the Secchi disk in the natural environment and avoids the problems of time-consuming and unstable observation by the naked eye. Moreover, this method also gives a water gauge recognition algorithm based on DeepLabv3+ to assist in measuring water transparency, which can control the error of water gauge recognition to about 1 cm. The relevant experimental results show that, compared with personal measurement, this method is more accurate, more objective, faster, and less costly.

## 2. Description of Algorithm

The overall framework of the algorithm is shown in Figure 1, including the following three parts: video pre-processing, determination of the critical position of the Secchi disk, water gauge recognition, and water gauge scale calculation.

### 2.1. Determination of the Critical Position of the Secchi Disk

A general camera is applied to imitate the action of personal measurement by using the Secchi disk and take some video of the disk and corresponding water gauge. The following pre-processing is done for the video of Secchi disk and the video of the water gauge: every three frames extract a picture and store in a fixed folder, respectively; the average value of the last 10 frames in the video is taken as the background image.

#### 2.1.1. Initial Image Crop of White Part of the Secchi Disk

The flow chart of the initial image crop of the white part of the Secchi disk is shown in Figure 2. First, Faster RCNN is used to recognize the first image, which contains the Secchi disk, and a rectangle, which can just surround the Secchi disk when it is obtained. Then, subtract the background image from all the images in the folder to get the image after subtracting the background. The position of the rectangular box with high average brightness is the position of the Secchi disk. Finally, crop the Secchi disk from the original image and determine the crop threshold of the white part by using the OTSU algorithm. The flow is as follows:(1)The brightness value of the whole picture is recorded as set C, and it is divided into two categories, one is recorded as set C1, the other is recorded as set C2, and C1⋂C2 = 0 and C1⋃C2 = C.(2)Take the brightness value K and put all the brightness values in the range of [0, k − 1] in set C1, and put the rest in set C2. The average value of the brightness value in set C1 is denoted as M1, and the proportion of the number of elements in set C1 to the number of elements in set C is denoted as P1; the average value of the brightness in set C2 is M2, and the proportion of the number of elements in set C2 to the number of elements in set C is P2. The mean value of the brightness in set C is recorded as m, and the formula for calculating the variance between classes is: g = P1 × (M1 − M)^2^ + P2 × (M2 − M)^2^.(3)The brightness value K is selected from 0 to 255 one by one, and the corresponding interclass variance is calculated every time. The K value corresponding to the maximum interclass variance divided by 255 is the final threshold. The brightness value that is greater than the threshold value is retained, and the rest are removed so that the white part on the Secchi disk can be cropped.

#### 2.1.2. Fine Image Crop of the White Part of the Secchi Disk

When the image of the Secchi disk is very blurred, the position and crop threshold of the Secchi disk cannot be determined according to the previous method. At this time, the image crop threshold will obviously deviate from all the previous threshold change trends, resulting in a jump. It is necessary to determine in which image the threshold jump happens first. The procedure is as follows: firstly, the crop threshold of the adjacent images is calculated according to the order of images. Secondly, the threshold difference of the adjacent images is calculated, and the absolute value of the difference is obtained. Next, the kmeans function is used to divide these differences into two categories. Finally, the average value of each category is calculated to determine in which image the threshold jump happens.

In this paper, the linear fitting method is used to determine the threshold value when the Secchi disk is blurred. While cropping the Secchi disk, the background of the corresponding position is also cropped. By normalizing the background brightness, the background brightness can be regarded as a Gaussian distribution, and the mean value is u and the standard deviation is σ. According to the rule of Gaussian distribution, about 98% of the background brightness values are less than u + 2σ. When the threshold value predicted by the linear fitting curve is less than u + 2σ, u + 2σ is used as the threshold value to prevent the water surface from being cropped as the white part of the Secchi disk.

#### 2.1.3. Determination of the Critical Position of the Secchi Disk by CNN

The crop results of the white part of the Secchi disk are divided into two categories by using a CNN classification network based on resnet18. One category indicates the existence of the Secchi disk, and the other category indicates that there is no Secchi disk. When the output of the classification network changes, it is the critical position of the Secchi disk.

### 2.2. Water Gauge Recognition and Water Level Calculation

#### 2.2.1. Water Gauge Segmentation

The flow chart of water gauge segmentation is shown in Figure 3. Firstly, the Deeplabv3+ algorithm is used to segment the corresponding water gauge image at the critical position of the Secchi disk. Since the water gauge in the image may be tilted, it is necessary to carry out tilt correction. The slope of the left or right edge of the water gauge can be calculated by Equation (1):(1)$$k=\frac{n\Sigma x_{i}y_{\dot{i}}-\Sigma x_{i}\Sigma y_{i}}{n\Sigma x_{i}^{2}-\Sigma x_{i}\Sigma x_{i}}$$
where *k* is the slope of the left or right edge of the water gauge, *x_i_* and *y_i_*, are the coordinate of the point on the left or right edge of the water gauge, and 1 ≤ *i* ≤ *n*, *n* is a positive integer.

According to the calculated slope, the slope of the water gauge image and the output of Deeplabv3+ are corrected at the same time, and then the water gauge can be segmented from the image.

#### 2.2.2. Characters Segmentation and Classification

There are two types of digital characters on the water gauge, one is located in the entire ten scale position, which is a relatively large character, and the other is located in the unit centimeter scale position, which is a relatively small character. Small number characters are difficult to segment and recognize, and this paper mainly segments the large number characters.

The flow chart of character segmentation is shown in Figure 4. The segmented water gauge image is transposed and mirrored to make it horizontal. The image is then binarized, reversed, and corroded, and each character is marked with a rectangular box. After the above operation, the small number characters are corroded or remain on the water gauge, while the large number characters still exist. In order to further distinguish large characters from other characters, the following method is designed: firstly, we calculated the area of each character’s rectangle; secondly, we divided these areas into two categories; next, we calculated the average area of each category—the corresponding character of the category with the larger average is the large character—and subsequently segmented the large character according to the position of the rectangle; and finally, the large characters are classified by CNN classification network based on resnet18.

#### 2.2.3. Water Gauge Scale Calculation

The position of each entire ten scale of the horizontal water gauge lies between each non-zero-digit character and the digit-zero character next to its right. For example, the position of scale 60 is between character 6 and character 0 to the right of it. The formula for calculating the position of each entire ten scale on the water gauge is shown in Equation (2):(2)$$x\left(k\right)=\frac{x{\_}_{left\left(k\right)}+x{\_}_{right\left(k\right)}}{2}$$
where *k* is a positive integer, *x*(*k*) is the position of the entire ten scale corresponding to the number *k* on the water gauge, *x*__*right*(*k*)_ is the right edge of the number *k*, and *x*__*left*(*k*)_ is the left edge of the number 0 to the right of the number *k*.

The reading scale calculation formula of the water gauge is obtained based on the ratio relationship, as shown in (3):(3)$$V\_reading=10\times \left(k-\frac{x\left(k\right)}{x\left(k+1\right)-x\left(k\right)}\right)$$
where *V_reading* is the reading value of the water gauge scale in cm.

When the Secchi disk is at the critical position, the corresponding water gauge reading value is the transparency.

## 3. Experiments

All the experiments are carried out on MATLAB 2020b. All the codes are implemented in the MATLAB programming language. The networks used in the experiment have very good performance. They are highly cited on Google and have been tested by thousands of researchers. Our experimental results also verify the excellent performance of these networks. The images are annotated with the Image Labeler tool on MATLAB 2020b. The training data is augmented with the albumentations toolbox or ImageDataAugmenter function. The neural networks are created and trained with the functions from the Deep Learning Toolbox on MATLAB 2020b. The training is carried out on a single NVIDIA GeForce RTX 2080Ti (11 GB).

### 3.1. Training of Neural Network in Determining the Critical Position of the Secchi Disk

The training of the neural network includes the training of the target detection network based on the Secchi disk and the training of the Secchi disk classification network.

More than 1000 pictures containing the Secchi disk are collected and annotated with the Image Labeler function. These images are augmented with the albumentations toolbox. The augmentation flow is as follows: each picture has a probability of 0.5 to be selected, 3 to 8 functions are randomly selected from albumentations to enhance the selected picture, and this operation is repeated several times. The functions used in the augmentation process and the number of data augmented with this function are shown in Table 1. After augmentation, there are 3069 images to train the target detection network based on the Secchi disk. SGDM [49,50] is selected as the optimization algorithm, the initial learning rate is 0.001, the learning rate attenuation mode is set to the initial learning rate, every two epochs are multiplied by 0.9, a total of 40 epochs are trained, and the minibatchsize is 16. With Alexnet [51] as the backbone, a Faster RCNN network model for network training is built. The loss curve and accuracy curve in the training process are shown in Figure 5.

Moreover, a total of 2000 images are collected to train the Secchi disk classification network, and, among them, 915 images show that there is no Secchi disk and 1085 images show that there is a Secchi disk. Eighty percent of the data are used as the training set, and the remaining data are used as the test set. The ImageDataAugmenter function is used to enhance the training data. The training data is enhanced online by randomly rotating it 90°, random mirroring, random horizontal translation between [−10, 10], and random vertical translation between [−10, 10]. SGDM is selected for the optimization algorithm, the minibatchsize is 32, a total of 25 epochs are trained, the initial learning rate is 0.001, the learning rate attenuation mode is set to the initial learning rate, and every epoch is multiplied by 0.92. The classification network is resnet18. A dropout layer is used to prevent overfitting. The loss and accuracy curve from the training process are shown in Figure 6.

### 3.2. The Test Results of the Critical Position of the Secchi Disk

#### 3.2.1. The Test Results of the White Part Crop of the Secchi Disk

The results of the initial crop and fine crop of the white part of the Secchi disk are shown in Table 2. As can be seen from Table 2, the effect of fine cropping is much better than that of initial cropping. When the Secchi disk is very blurred, the position of the Secchi disk cannot be determined by initial cropping, so the crop result is an irregular image. However, fine cropping can determine the location of the Secchi disk and crop the white part of the Secchi disk. When the Secchi disk is completely invisible, it is easy for the initial crop to crop the water surface, but the fine crop will not.

#### 3.2.2. Test Results of Critical Position Determination by Classification Network

The accuracy of the Secchi disk classification network on the test set is 100.00%, which meets the requirements. After pre-processing the several collected videos, the previous crop algorithm of the white part of the Secchi disk is used to crop the videos. The crop results are sent into the trained classification network, and the image in which the Secchi disk is just invisible is determined according to the classification results. Some experimental results are shown in Table 3. The test results show that the critical position of the Secchi disk is basically accurate, and the Secchi disk is almost invisible in the determined image.

### 3.3. Training of Neural Network in Water Gauge Recognition

#### 3.3.1. Deeplabv3+ Network Training

1100 images containing the water gauge are collected for the training set. The training data is augmented with the albumentations toolbox. The augmentation flow is as follows: each image has a probability of 0.6 to be selected, 2 to 8 functions are randomly selected from albumentations to enhance the selected image, and this operation is repeated several times. The functions used in the augmentation process and the number of data augmented with this function are shown in Table 4. After augmentation, the training set contains 5066 images. Another 200 images are applied as the test set. The Image Labeler function is used for data annotation. The optimization algorithm selects Adam [49,50], the initial learning rate is set to 0.002, the learning rate attenuation mode is set to the initial learning rate and is multiplied by 0.5 every 10 epochs, and the minibatchsize is set to 8. A total of 60 epochs are trained. Resnet18, Xception [52], and Mobilenetv2 [53] were taken as backbones, respectively, and the Deeplabv3+ network model is built for network training. The loss and accuracy curve from the training process are shown in Figure 7.

#### 3.3.2. Training of Character Classification Network

More than 10,000 digital character images are collected. Eighty percent of the data are used as the training set, and the rest are used as the test set. The ImageDataAugmenter function is used to enhance the training data. The training images are enhanced online by randomly rotating them 10°, random horizontal translation between [−10, 10], and random vertical translation between [−10, 10]. SGDM is selected for the optimization algorithm, the minibatchsize is set to 32, a total of 25 epochs are trained, the initial learning rate is set to 0.001, and the learning rate attenuation mode is set to the initial learning rate and is multiplied by 0.9 every epoch. The classification network is resnet18. The dropout layer is used to prevent overfitting. The loss and accuracy curve from the training process are shown in Figure 8.

### 3.4. Experimental Results of Water Gauge Recognition

The main evaluation indexes of the Deeplabv3+ semantic segmentation effect are pixel accuracy (PA) and mean intersection over union (MIoU). The calculation formula is shown in (4) and (5):(4)$$\mathrm{PA}=\frac{\sum_{i=0}^{k}\;P_{ii}}{\;\sum_{i=0}^{k}P_{j}\sum_{j=0}^{k}P_{ij}}$$
(5)$$\mathrm{MIoU}=\frac{1}{k+1}\sum_{i=0}^{k}\;\frac{P_{ii}}{\sum_{j=0}^{k}\;P_{ji}+\sum_{j=0}^{k}P_{ji}-P_{ii}}$$
where *P_ij_* is the part belonging to category *j* among the pixels predicted by category *i*.

The larger the MIoU, the better the semantic segmentation. Table 5 shows the performance of Deeplabv3+ on the test set under different backbones. As can be seen from Table 5, the PA of different backbones is close with each other. MIoU is more reliable than PA in this case. Resnet18 is faster than other backbones, but Mobilenetv2 has the highest MIoU.

The accuracy of the water gauge character classification network in the test set is 99.98%, which basically meets the requirements. The calculation of the water gauge reading is tested. Some test results are shown in Table 6, and the error is about 1 cm.

### 3.5. Overall Test Results of the Algorithm

Some video of the Secchi disk and the corresponding water gauge video are collected, and a graphical user interface (GUI) is designed to test the overall algorithm. Some results are shown in Table 7. From the test results, it can be seen that the transparency of this algorithm is higher than that of personal eyes, and the maximum error is 3.4 cm. Considering that personal eye observation is easily affected by subjective experience, and personal eye image resolution is weaker than that of the computer, the result of this algorithm is higher than that of personal eye observation in theory. Within an allowable error range, the test results of this algorithm are reliable, and even closer to the real value than the manual observation results (the manual observation value refers to the weighted average value obtained after multiple measurements by multiple observers).

## 4. Conclusions

This paper presents an algorithm for transparency measurement based on computer vision. The measurement method of water quality transparency based on Secchi disk includes the following two aspects: blurred Secchi disk detection and the depth of Secchi disk detection. A variety of image processing and depth learning methods are used in this algorithm. Image processing is used to crop the white part of the Secchi disk, the classification network based on resnet18 is applied to classify the segmentation results and determine the critical position of the Secchi disk, and the Deeplabv3+ network is used to segment the water gauge and the characters on the water gauge. The algorithm can more accurately determine the critical position of the Secchi disk and obtain more exact and objective water quality transparency data. The overall test results from the algorithm are higher than that of personal observation. This technology has strong practical value, and it is more accurate and objective and less time-consuming than that of personal observation (the results of personal measurement are easily affected by subjective experience and objective environment). It can even help to form a unified standard of water quality transparency based on Secchi disks in the future. The application of these artificial intelligence technologies, such as deep learning, in water quality monitoring is very promising.

## Figures and Tables

**Figure 1 sensors-22-05399-f001:**
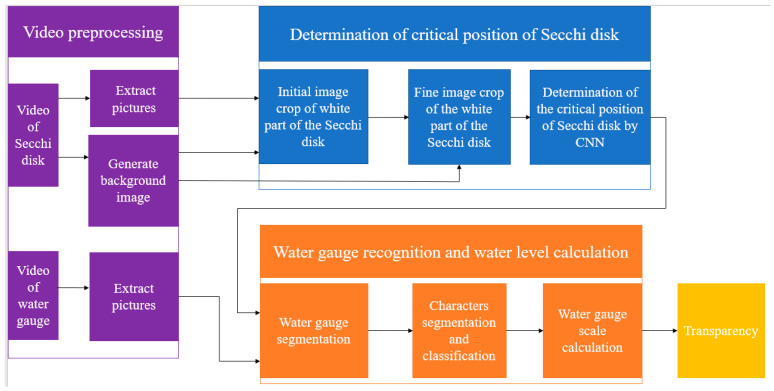
Overall framework of the algorithm.

**Figure 2 sensors-22-05399-f002:**
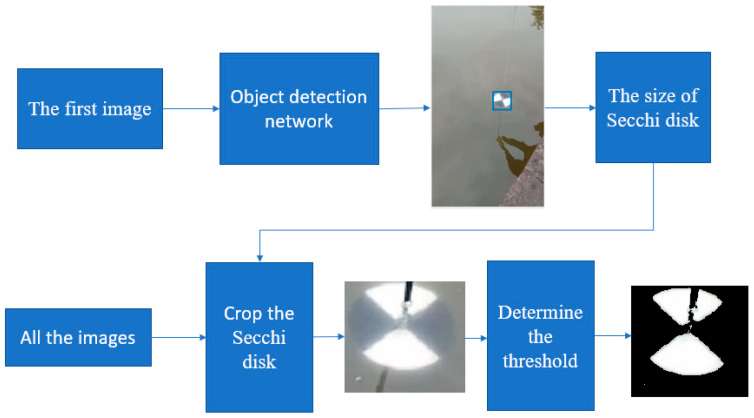
Flow chart of initial image crop of the white part of the Secchi disk.

**Figure 3 sensors-22-05399-f003:**
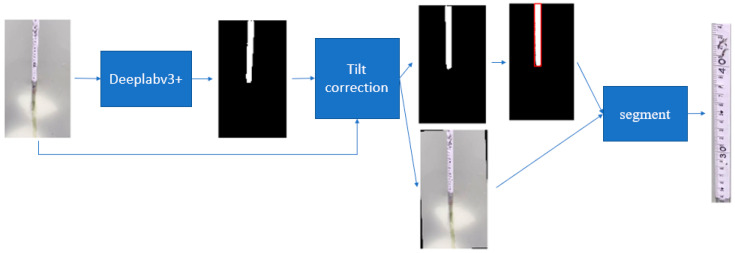
Flow of water gauge segmentation.

**Figure 4 sensors-22-05399-f004:**
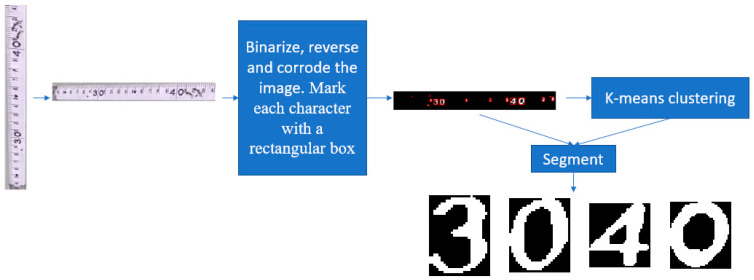
Flow chart of character segmentation.

**Figure 5 sensors-22-05399-f005:**
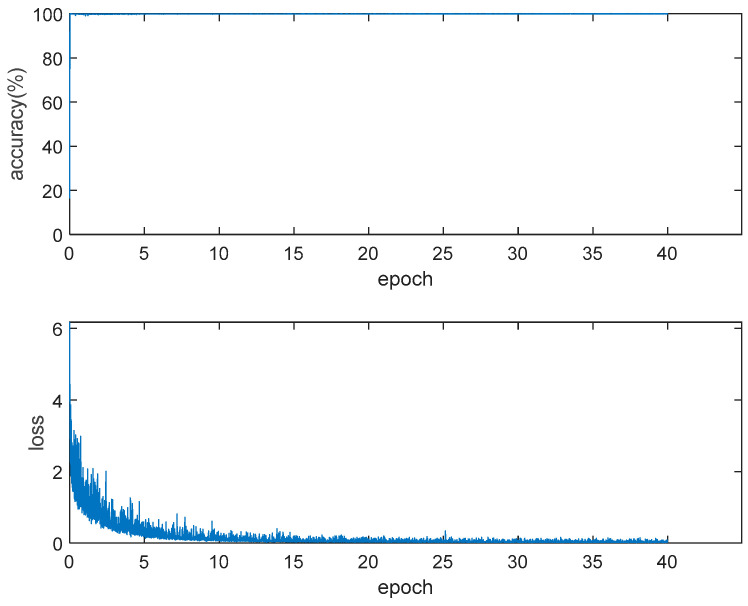
Loss and accuracy in Faster RCNN training.

**Figure 6 sensors-22-05399-f006:**
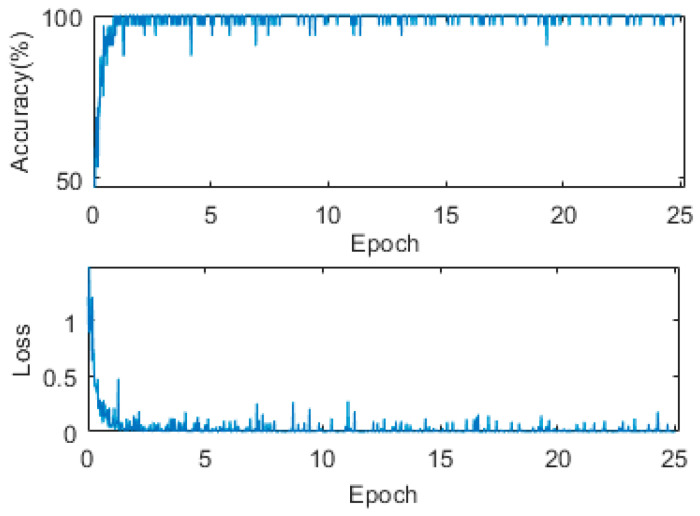
Loss and accuracy in classification network training of Secchi disk.

**Figure 7 sensors-22-05399-f007:**
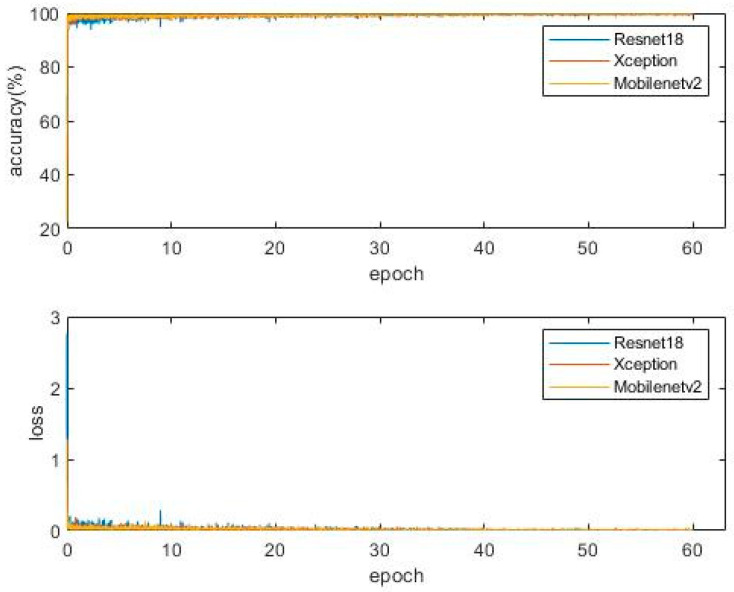
The loss and accuracy curve from the training process.

**Figure 8 sensors-22-05399-f008:**
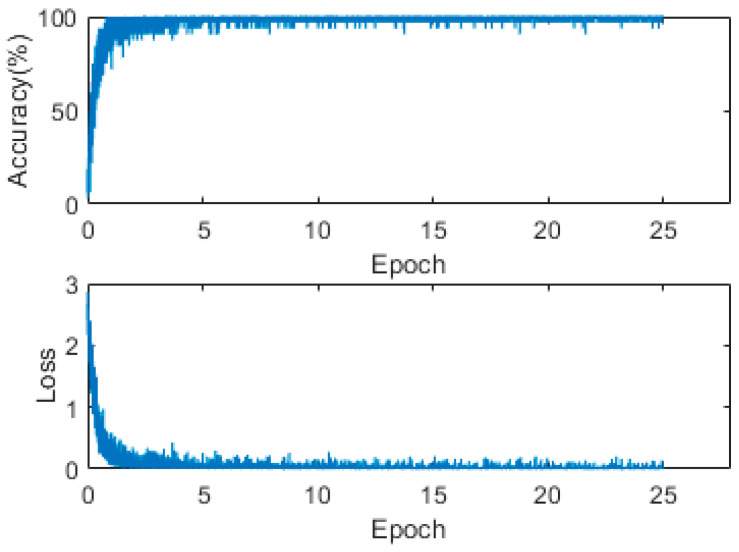
Loss and accuracy curve from character classification network training.

**Table 1 sensors-22-05399-t001:** Functions used for training data augmentation of Faster RCNN network.

Name of the Function	Role of the Function	Number of Data Augmented with the Function
MedianBlur	Blur the image using a median filter	460
Cutout	random dropout of the square region in the image	433
RandomSunFlare	Simulate sun flare for the image	446
RandomFog	Simulate fog for the image	470
RandomRain	Simulate rain for the image	437
MotionBlur	Apply motion blur to the image	447
GlassBlur	Apply glass blur to the image	481
Superpixels	Transform images to their super-pixel representation	476
Sharpen	Sharpen the image	464
ImageCompression	Decrease jpeg compression of the image	382
MultiplicativeNoise	Multiply image to random number	462
CLAHE	Apply contrast limited adaptive histogram equalization to the image	474
HorizontalFlip	Flip the image horizontally	438
Rotate	Randomly rotate the image	471
VerticalFlip	Flip the image vertically	429
RandomCrop	Randomly crop the image	461
ShiftScaleRoate	Randomly apply affine transforms	426
Perspective	Randomly apply perspective transforms	436
RandomSnow	Simulate snow for the image	417
HueSaturationValue	Randomly change the hue, saturation, and value of the image	485
ISONoise	Apply camera sensor noise	439
GaussNoise	Apply Gaussian noise	428

**Table 2 sensors-22-05399-t002:** The results from cropping the white part of the Secchi disk.

Image	Blur Degree of Secchi Disk	Initial Crop	Fine Crop
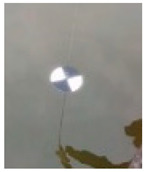	clear	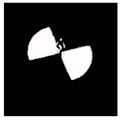	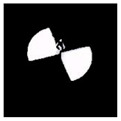
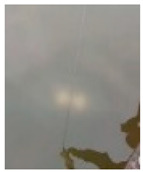	blurred	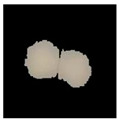	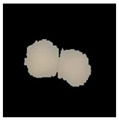
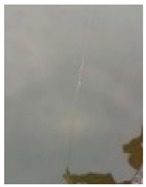	very blurred	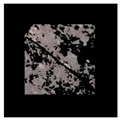	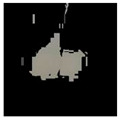
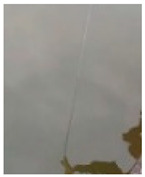	completely invisible	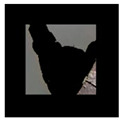	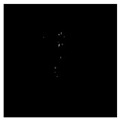

**Table 3 sensors-22-05399-t003:** Test results of critical position determination by classification network.

Video	Initial Position	Completely Invisible Position	Critical Position
1	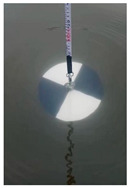	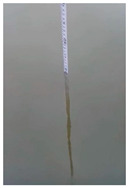	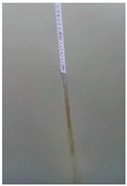
2	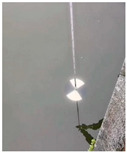	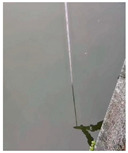	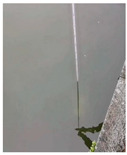
3	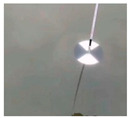	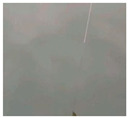	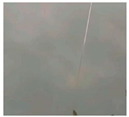
4	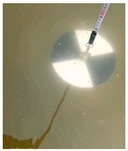	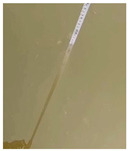	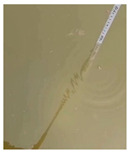
5	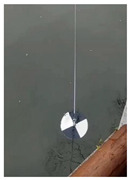	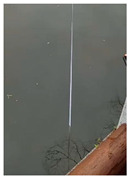	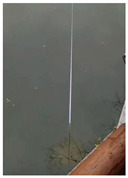

**Table 4 sensors-22-05399-t004:** Functions used for training data augmentation of Deeplabv3+ network.

Name of the Function	Role of the Function	Number of Data Augmented with the Function
MedianBlur	Blur the image using a median filter	691
Cutout	random dropout of the square region in the image	689
RandomSunFlare	Simulate sun flare for the image	630
RandomFog	Simulate fog for the image	675
RandomRain	Simulate rain for the image	663
MotionBlur	Apply motion blur to the image	719
GlassBlur	Apply glass blur to the image	670
Superpixels	Transform images to their super-pixel representation	672
Sharpen	Sharpen the image	649
RandomShadow	Simulate shadow for the image	655
MultiplicativeNoise	Multiply image to random number	701
CLAHE	Apply contrast limited adaptive histogram equalization to the image	654
HueSaturationValue	Randomly change the hue, saturation, and value of the image	666
RandomBrightnessContrast	Randomly change brightness and contrast of the image	657
RandomSnow	Simulate snow for the image	686
GaussianBlur	Apply Gaussian blur to the image	650
Emboss	Emboss the image	661
GridDropout	Drop out square region of the image in grid fashion	674
ImageCompression	Decrease jpeg compression of the image	656
ISONoise	Apply camera sensor noise	681
HorizontalFlip	Flip the image horizontally	652
Rotate	Randomly rotate the image	650
VerticalFlip	Flip the image vertically	672
RandomCrop	Randomly crop the image	649
ShiftScaleRotate	Randomly apply affine transforms	626
Perspective	Randomly apply perspective transforms	680

**Table 5 sensors-22-05399-t005:** Performance of Deeplabv3+ in different models.

Backbone	Image Size	FPS	PA	MIoU
Resnet18	640 × 480	23.95	99.55%	94.20%
Xception	640 × 480	16.97	99.57%	94.38%
Mobilenetv2	640 × 480	17.35	99.57%	94.48%

**Table 6 sensors-22-05399-t006:** Water gauge reading calculation test results.

Input Image	Water Gauge after Segmentation	Ground Truth (cm)	Measurement (cm)
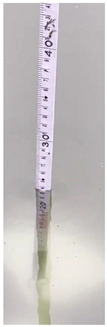	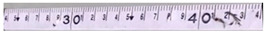	23.6	23.8
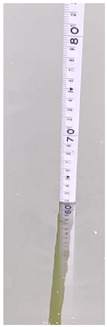	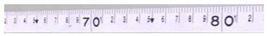	60.8	61.9
	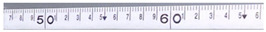	46.0	46.4
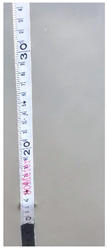	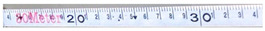	13.2	13.8
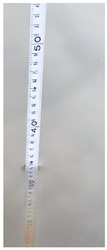	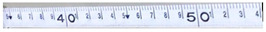	34.4	34.5

**Table 7 sensors-22-05399-t007:** Overall test results of the algorithm.

Video	Initial Position of Secchi Disk	Critical Position of Secchi Disk	Transparency of Manual Measurement	Transparency of GUI Measurement
1	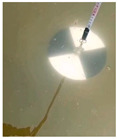	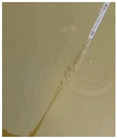	55.0 cm	57.8 cm
2	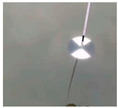	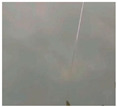	58.0 cm	61.4 cm
3	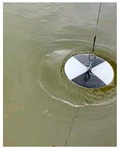	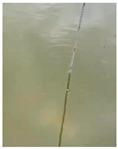	50.0 cm	52.7 cm
4	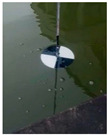	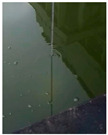	40.0 cm	42.5 cm

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
