# Peer review of "Water Quality Measurement and Modelling Based on Deep Learning Techniques: Case Study for the Parameter of Secchi Disk"

_sensors, 2022, doi:10.3390/s22145399_

Round 1
Reviewer 1 Report
Summary
The authors used deep learning and image processing methods to obtain Secchi disk measurements. A comparison between the proposed method and manual measurements of Secchi disk readings is presented. An in-detail methodology flowchart is provided along with relevant explanation. The proposed techniques are useful when implemented for long-term water-quality monitoring stations. In addition to Secchi disk, there are many water quality sensors for measuring transparency that are available and regularly used for monitoring purposes worldwide. Following are some of the comments that need to be addressed before publication.
Comments
1. Please provide more applications of water transparency mentioned in lines 31 along with references. For example, primary production of phytoplankton, seagrasses health, coral reefs
2. I do not understand lines 31-32, what does daily production mean? is it daily production of various organisms or plants in water? Or is it phytoplankton? Please be specific and provide relevant references.
3. Line 34: Please rephrase “It is a black alternating with white disk”.
4. Lines 48-49: Rephrase “…transparency of large area water area..”
5. Line 49-50: I do not completely agree with these lines. There are a lot of advantages in using spectral images by satellites. Most of the ocean color satellites acquire images over large areas and algorithms are in development to derive various biogeochemical parameters. In comparison with the cost of field measurements to acquire various parameters, satellite derived parameters are cost effective. The accuracy of the derived parameters however depends on the area of implementation, clear ocean, coastal, inland, estuarine and turbid waters.
6. Line 50-51: There is no connection here. Why did the authors switch directly from using satellite images to machine learning? Machine learning is not an alternative to satellite imagery. They are tools to enhance our understanding of the complex patterns. Please provide more information if machine learning is a way to improve the information obtained from satellite images or its better performance over traditional mathematical models. I see that the next few lines indicate some information about various studies using ML to predict water quality. However there is a disconnect here. Please provide more information.
7. Line 53: Where did the author Sentas use the ANN model?
8. Line 54: Same as above, provide more information on the previous literature.
9. Please replace the word “researches” with “studies” in the entire document.
10. Line 57: What are other parameters, please rephrase.
11. Line 80: It should be noted that deep learning is not an image processing technique. DL is used for many applications and image processing is one of them. Please rephrase accordingly.
12. Line 81: Provide the specifications of the camera
13. Line 86: Provide full form of OTSU.
14. While the results of the proposed algorithm looks promising, a comparison of the proposed NN’s with other algorithms would be beneficial.
15. More information on why the authors have chosen resnet, Deeplabv3+ and Faster RCNN are chosen over other methods need to be provided. What are the advantages of the chosen techniques over other methods. Please elaborate.
16. Have the authors tried to implement other NNs apart from the chosen ones mentioned above?
17. Following references regarding the Secchi disk theory are substantially important for research carried out using Secchi disk depth.
Lee, Z., Shang, S., Hu, C., Du, K., Weidemann, A., Hou, W., Lin, J., Lin, G., 2015. Secchi disk depth: A new theory and mechanistic model for underwater visibility. Remote Sens. Environ. 169, 139–149. https://doi.org/https://doi.org/10.1016/j.rse.2015.08.002
WERNAND, M. R.. On the history of the Secchi disc. Journal of the European Optical Society - Rapid publications, Europe, v. 5, apr. 2010. ISSN 1990-2573. Available at: <http://www.jeos.org/index.php/jeos_rp/article/view/321>. Date accessed: 03 Jul. 2022. doi:199.
Aas, E., Høkedal, J., and Sørensen, K.: Secchi depth in the Oslofjord–Skagerrak area: theory, experiments and relationships to other quantities, Ocean Sci., 10, 177–199, https://doi.org/10.5194/os-10-177-2014, 2014.

Reviewer 2 Report
According to my knowledge, Secchi disk depth is a measure of water transparency, which has been explained with the underwater visibility theory. In recent years, the remote sensing image was used for estimating the Secchi disk depth of water at a large scale. However, this work used deep learning to extract the critical position of Secchi disk based on a video about the measurement process. However, the motivation of this research seems to propose a method that can automatically obtain the depth of a certain Secchi disk in a local area. However, during the measurement process, the depth of a Secchi disk can be recorded by the person who videotaped. And, the manual measurement is more accurate and time-saving than the machine learning approaches. The authors should explain the the merits and necessity of this work.
Round 2
Reviewer 2 Report
The authors have addressed my initial concerns and recommendations. I suggest that, the main contribution and novelty of this work should be listed in the Introduction.
Author Response
Dear Prof. Dr. Reviewer:
Thank you for your letter and for the reviewer’s comments concerning our manuscript entitled “Water Quality Measurement and Modelling Based on Deep Learning Techniques: Case Study for the Parameter of Secchi Disk” (ID: sensors-1800626R2). The comments are great significance and helpful to us:
We have studied comments carefully and have made correction which we hope meet with approval. Minor revised portion are marked in other color at the end of the Introduction of the paper (Line124-133 of new version). The main corrections in the paper and the responds to the reviewer’s comments are as following:
The authors have addressed my initial concerns and recommendations. I suggest that, the main contribution and novelty of this work should be listed in the Introduction.
Response to comment: Minor revised portion are marked in other color at the end of the Introduction of the paper (Line124-133 of new version). The details are as follows:” Basically, the main contribution and novelty of this work are as follows: creatively proposes an algorithm based on deep learning and image processing technology to measure the transparency of water quality with Secchi disk. The algorithm adopts a method to determine the critical position of the Secchi disk, which can accurately determine the critical position of the Secchi disk in the natural environment, and avoids the problems of long time-consuming and unstable observation by naked eyes. Moreover, this method also gives a water gauge recognition algorithm based on DeepLabv3+ to assist in measuring water transparency, which can control the error of water gauge recognition to about 1cm. Relevant experiment results show that compared with personal measurement, this method is more accurate, more objective, faster and less costly.”
Special thanks to you for your good suggestion.
Submission Date: 16 July 2022